# CREPE (CREate Primers and Evaluate): A Computational Tool for Large-Scale Primer Design and Specificity Analysis

**DOI:** 10.3390/genes16091062

**Published:** 2025-09-10

**Authors:** Jonathan W. Pitsch, Sara A. Wirth, Nicole T. Costantino, Josh Mejia, Rose M. Doss, Ava V. A. Warren, Jack Ustanik, Xiaoxu Yang, Martin W. Breuss

**Affiliations:** 1Department of Pediatrics, Section of Genetics and Metabolism, University of Colorado School of Medicine, Aurora, CO 80045, USA; 2Department of Human Genetics, University of Utah, Salt Lake City, UT 84112, USA; xiaoxu.yang@genetics.utah.edu

**Keywords:** primer design, targeted amplicon sequencing, polymerase chain reaction, bioinformatic tool

## Abstract

Background/Objectives: Polymerase chain reaction (PCR) is ubiquitous in biological research labs, as it is a fast, flexible, and cost-effective technique to amplify a DNA region of interest. However, manual primer design can be an error-prone and time-consuming process depending on the number and composition of target sites. While Primer3 has emerged as an accessible tool to solve some of these issues, additional computational pipelines are required for appropriate scaling. Moreover, this does not replace the manual confirmation of primer specificity (i.e., the assessment of off-targets). Methods: To overcome the challenges of large-scale primer design, we fused the functionality of Primer3 and In-Silico PCR (ISPCR); this integrated pipeline, CREPE (CREate Primers and Evaluate), performs primer design and specificity analysis through a custom evaluation script for any given number of target sites at scale. Results: CREPE’s final output summarizes the lead forward and reverse primer pair for each target site, a measure of the likelihood of binding to off-targets, and additional information to aid decision-making. We provide this through a customized workflow for targeted amplicon sequencing (TAS) on a 150 bp paired-end Illumina platform. Experimental testing showed successful amplification for more than 90% of primers deemed acceptable by CREPE. Conclusions: We here provide CREPE, a software platform that allows for parallelized primer design for PCR applications and that is optimized for targeted amplicon sequencing.

## 1. Introduction

Since its inception in 1983, polymerase chain reaction (PCR) has become ubiquitous in biological research labs [1]. In particular, in genetics research, the ability to amplify a region of interest followed by sequence analysis has been revolutionary. Although technologies have evolved in many ways, PCRs still underlie many analytical pipelines. For instance, targeted amplicon sequencing (TAS) and various derivative methods rely on separate PCRs for initial amplification before employing next-generation sequencing for pooled analysis [2,3,4,5]. While classical Sanger sequencing requires the assessment of each PCR in isolation, those employing next-generation sequencing allow for the parallel analysis of sequences and are inherently scalable.

Integral to PCR is the design of appropriate primers that amplify a region of interest with high specificity. Largely ‘manual’ primer design with assisting technologies that assess primer features (e.g., melting temperature, GC-content, or predicted hairpin structures) is still employed in many laboratories. However, for most applications, the automation of this process provided by tools like Primer3 is superior, especially when considering scaling [6,7]. Primer3 has become an important standard in the scientific community, as it is accessible through a graphical user interface (GUI) [8]. It can also be deployed from the command line, which allows for the scaling of primer design even with a basic level of computational skills. In either configuration, it performs an analysis of possible primers to determine their viability for PCR using standard metrics that can be modified by the user [6,7]. We and others have successfully employed this approach to scale primer design across tens to hundreds of loci of interest [2,5,9].

Yet this does not remove the necessity of an additional manual review of each designed primer pair for many applications in genomics and genetics: any designed primers may also bind to other regions in the genome in addition to the intended target—sites commonly referred to as ‘off-targets’ [10]. Thus, primers are typically reviewed and selected for their specificity to the target site. This is generally achieved through a manual review of off-target analyses provided through tools such as the Primer Basic Local Alignment Search Tool (Primer-BLAST, https://www.ncbi.nlm.nih.gov/tools/primer-blast/index.cgi?LINK_LOC=BlastHome accessed on 9 September 2025) or in-Silico PCR (ISPCR) [11,12]. While Primer-BLAST provides a very powerful GUI and reports useful metrics to assess potential off-targets, it is currently not compatible with locally run batched analyses. While BLAST itself can be run locally via the command line, the setup and usage are complex, and the output analysis is missing information that may aid in overall experimental design, and it can be acquired with other tools [13]. ISPCR, on the other hand, can be deployed from the command line and allows for the required scaling. While its default settings are geared towards the detection of perfect off-target matches for the input primer pair, changes to the settings of the underlying algorithm behind ISPCR, the BLAST-Like Alignment Tool (BLAT) enables the identification of imperfect off-target matches that might also result in aberrant PCR products in practice [14].

Here, we present a novel computational primer design tool that takes advantage of the properties of both Primer3 and ISPCR: CREPE (CREate Primers and Evaluate) creates primer pairs for any number of input target sites and performs specificity analysis with ISPCR. A downstream analysis evaluation script further refines and summarizes these results, which generates informative annotations for the primers of each target site. By merging primer design and advanced specificity analysis into a single tool, CREPE is a simple solution for researchers performing large-scale PCR experiments. In contrast to more complex tools that can allow for comparable functionality [15,16,17], our proposed solution is streamlined for the specific design of parallel PCRs. Consequently, we provide a default setup to optimize the primer design process for TAS experiments analyzed on a 150 bp paired-end sequencing platform. To increase yield for this application, we include the iterative design of alternative amplicons that are compatible to this type of analysis.

## 2. Methods

### 2.1. CREPE Pipeline Software Versions

The current version of the CREPE pipeline, including sample input and output files, is available at https://github.com/martinbreuss/BreussLabPublic/tree/main/CREPE (accessed on 9 September 2025). Up-to-date software version requirements and download and installation instructions can be found there. For the here-employed CREPE v1.02, the following versions of essential tools were used: Bedtools v2.26; Biopython v1.79; ISPCR v33; Primer3 v2.6.1; Python v3.7.7; Pysam v0.15.4; and Pandas v1.3.5 [6,7,12,18,19,20,21].

### 2.2. Run Time and Local Storage Testing

The run time and storage testing data were established using an instance of CREPE installed on an M1 Apple iMac with 16 GB local memory (Apple, Cupertino, CA, USA).

### 2.3. CREPE Pipeline Primer3 and ISPCR Overview

In short, a customized input file (see Appendix A) with the required columns ‘CHROM’, ‘POS’, and ‘PROJ’ was processed using Python to generate a machine-readable input file for Primer3 (see Appendix A); in parallel, a genome reference file as used to retrieve local sequence information. Consequently, the chromosome and positions of the target region needed to be compatible with the reference file (UCSC’s GRCh38.p14 as default). The generated primer pairs, including forward–forward and reverse–reverse primer pairs for each target site, were then formatted for input into ISPCR with the following algorithm parameters: **-minPerfect = 1** (minimum size of perfect match at 3′ end of primer), **-minGood = 15** (minimum size where there must be two matches for each mismatch), **-tileSize = 11** (the size of match that triggers an alignment), **-stepSize = 5** (spacing between tiles), and **-maxSize = 800** (maximum size of PCR product) [12]. A FASTA file containing alignment information, primer ID, the forward and reverse primer sequences, and the amplicon sequence for each primer pair was then written by ISPCR. Additionally, a BED file containing the chromosome, amplicon start position, amplicon end position, and score was written by ISPCR for the generated primer pairs. The score calculated by ISPCR came from an analysis of primer mismatches and the resulting viability of PCR. On-target primer pairs with no mismatches had a score of 1000.

### 2.4. CREPE Pipeline Evaluation Script (Off-Target Assessment)

The FASTA and BED files from ISPCR were read by a custom Python script. Primer pairs aligning to decoy contigs in the reference genome were removed. The information from ISPCR and Primer3 for the remaining primer pairs was formatted into a single output file. To eliminate extremely low-quality off-targets found by ISPCR within the maximum amplicon size determined by the maxSize parameter (800 bp by default), any primer pair with a score less than 750 was arbitrarily filtered out. The amplicon sequence for each primer pair was parsed to identify the number and location of mismatches in the forward and reverse primers. All off-target amplicons found for any given target site were aligned to the on-target amplicon and a normalized percent match to the on-target amplicon was calculated using $normalized\;\%\;match=\;\frac{alignment\;score}{len(amplicon)}\;$. The normalized percent match was calculated and recorded twice, first by dividing the alignment score (from Biopython PairwiseAligner) by the length of the off-target amplicon (normalized_match_to_test_amplicon); then, a second match was calculated by dividing the alignment score by the length of the on-target amplicon (normalized_match_to_gold_amplicon). In this way, the normalized match for off-target amplicons of any size was properly measured. Any remaining off-target amplicon with a normalized match percentage between 80 and 100% was considered a high-quality (concerning) off-target (HQ-Off). Any remaining off-target amplicon with a normalized match percentage less than 80% was considered a low-quality (non-concerning) off-target (LQ-Off).

### 2.5. CREPE Pipeline Output File Format

The results from the evaluation script (E-script) were merged onto the input CSV file, sorted by chromosome and position, and then written to a tab-delimited text file compatible with commonly used programming languages and spreadsheet editors. In addition to the columns provided in the input file, the output file contained a variety of columns, including **variant_ID** (PROJ_CHROM_POS, e.g., clinvar_1_ 944041), **Primer3** [Boolean] (True if a viable primer pair was created by Primer3), **ISPCR** [Boolean] (True if the primer pair was accepted by ISPCR), **primer name** (describes primer type, e.g., clinvar_1_944041_TAS-opt), **TAS-opt** [Boolean] (True if primer pair is a TAS-opt pair), **forward primer name** (clinvar_1_944041_TAS-opt_F), **forward primer sequence**, **forward primer melting temperature**, **reverse primer name** (clinvar_1_944041_TAS-opt_F), **reverse primer sequence**, **reverse primer melting temperature**, **amplicon start position**, **amplicon end position**, **amplicon length**, **primer count** (number of off-targets in addition to primer pair, e.g., 1 = no off-targets, 2 = one off-target, 3 = two off-targets, etc.), and **concerning off-targets** [Boolean] (True if primer pair has a high quality off-target).

### 2.6. Targeted Amplicon Sequencing

Target Selection and Primer Design: To test CREPE with TAS, 1000 variants were randomly selected from the 20240603 version of the ClinVar database to simulate potentially relevant clinical targets. No other filters were applied for these variants. The targeted sites were then used as input for our TAS-optimized CREPE v.1.02 pipeline.

Experimental Procedures: Overall, experimental procedures were performed as described previously [2,4,5]. For the selected subset, primers were ordered from a commercial provider (IDT), diluted to 5µM in nuclease-free water, and used for a PCR with GoTaq Master Mix (M7133, Promega) according to the manufacturer’s instructions (30 µL volume). A standard amplification protocol was employed on ‘control’ genomic DNA (100 °C lid temperature; cycles: 1 cycle 95C 3′; 40 cycles 95C 30″, 60C 20″, 72′C 20″; 1 cycle 72C 7′). The resulting amplicons were visualized using standard gel electrophoresis techniques. For further processing, PCR products were cleaned up enzymatically with exonuclease 1 (NEB M0293L) and shrimp alkaline phosphatase (NEB M0371S) (‘ExoSAP’) for at least 15 min according to the manufacturer’s instructions. The resulting amplicons were quantified using the Quant-iT double-stranded DNA high-sensitivity kit (Q33120, Thermo Fisher) on a fluorescent M200 plate reader (30050303, Tecan) according to the manufacturer’s instructions. Standards were read in duplicate, while samples were read in singlet. All PCRs were then pooled at equal quantities, with no pool having two overlapping amplicons. These pools were then processed and barcoded using the KAPA HyperPrep kit (KK8501-KK8505, Roche) and UDIs for Illumina Platforms (ILM 20023784, IDT) according to the manufacturer’s instructions. Processed pools were combined to achieve an equal representation of each amplicon at an overall final concentration of 10 nM in 30 µL for submission to the Genomics and Microarray Sequencing Core at CU Anschutz (Genomics Shared Resource RRID: SCR_021984). Sequencing was performed on an Illumina NovaSEQ X for paired-end sequencing (2 × 150 bp) at a minimum read depth of 50 M reads per sample.

Sequencing Data Processing: Paired-end FASTQ files were aligned to the GRCh38.p14 reference genome with BWA mem (0.7.15) and converted to a BAM file with Samtools view (1.9) [22,23]. BAM files were sorted and indexed with Samtools sort and index. Mate-pair information between mates was verified with Picard FixMateInformation (2.18.29) [24]. A read group was added to the sequencing data with Picard AddOrReplaceReadGroups. Lastly, base recalibration is performed with GATK BaseRecalibrator and ApplyBQSR (4.2.6.1) [25].

Separation of On-target Coverage and Background Enrichment: To determine primer pair efficiency, the sequencing data for each group was separated into two BAM files with Samtools view. One BAM file contained only reads that overlap with the target amplicons, while the other contained any other sites in the genome with a read depth greater than zero. In this way, the on-target coverage was separated from the background enrichment. Using the 95th percentile of the background read depth distribution as an estimate of background enrichment in each dataset, we identified primer pairs that did not have enough on-target coverage to be distinguishable from the background. The full processing pipeline is available at https://github.com/martinbreuss/BreussLabPublic/tree/main/CREPE/scripts (accessed on 9 September 2025).

### 2.7. TAS Analysis

We created pileup files for each target site with Samtools mpileup. Lastly, we employed a TAS analysis script to parse the pileup files, record the read depth at that site, calculate the mutant allelic fraction (MAF) of the target site, and determine the Wilson 95% confidence interval around that MAF [4]. In this way, a comprehensive measure of each primer pair’s performance was obtained.

### 2.8. Measuring Off-Target Coverage

To measure off-target coverage, we used Bedtools merge to combine overlapping off-target enrichment sites. This reduced the number of predicted off-targets to 400. As some predicted off-targets had large amplicons (>500 bp), it was essential to treat each merged region as a single off-target to accurately measure off-target coverage. Samtools view was then used to count the number of reads in each region.

### 2.9. Data Analysis and Visualization

All data analysis and visualization after CREPE and TAS were performed in a Jupyter notebook with Matplotlib v3.5.1, Numpy v1.21.5, Pandas v1.4.2, Python v3.9.12, Scipy v1.7.3, Seaborn v.0.11.2, and Statsmodel v0.13.2 [21,26,27,28,29,30].

### 2.10. Human Subject Statement

Samples were from a healthy 40-year-old male who consented under the IRB protocol 21-4071; these were considered as the ‘control’.

### 2.11. DNA Sample Extraction

DNA samples were extracted using a commercially available extraction kit according to the manufacturer’s instructions (Cat: 69506, Qiagen DNeasy Blood & Tissue Kit).

## 3. Results and Discussion

### 3.1. Overview of the Approach

At its core, CREPE works in three parts as follows: 1. primer design with Primer3; 2. specificity analysis with ISPCR; and 3. off-target evaluation with a custom Python evaluation script (E-script) (Figure 1A). The user input consists of a simple list of interrogated target sites, and the output is a human-readable list of designed primers with additional information to aid a user’s assessment of primer quality (see Section 2, the legend in Appendix A, and Appendix A for details). In addition to a summary contained in the main output file, Appendix A obtained from the primer evaluation for advanced users. Similarly, for less experienced users, we categorize primers into those that have no off-targets, those with low-quality off-targets (LQ-Off, i.e., amplification is likely but similarity to on-target amplicon is low based on the in silico analysis), and those with high-quality off-targets (HQ-Off, i.e., amplification is likely and similarity to on-target amplicon is high; see Section 2 for details).

As default, all parameters are currently optimized for TAS experiments and the deep sequencing of target sites to assess genomic mosaicism employing Illumina 150 bp paired-end sequencing [2,5]. These parameters for the desired amplicon size and the distance of the amplicon end from the target site can be directly accessed in the CREPE Python script as needed. To optimize computation, CREPE initially performs the outlined three steps for ‘TAS-optimized’ (TAS-opt) primer pairs that are predicted to create an ideal amplicon size for TAS (190 bp) and coverage of the desired target by both the forward and the reverse read. However, in our experience, Primer3 is typically unable to design a PCR primer pair for all target sites adhering to these strict size requirements. Yields can be improved by considering larger amplicons that only cover the target from one end of the read (Figure 1B). While this reduces the theoretical coverage by half, it enables the assessment of additional targets. In practice, to optimize run time, this is performed through an iterative process. Following the off-target evaluation for the TAS-opt primers, CREPE attempts to find additional primer pairs for the target sites that failed to create a TAS-optimized amplicon by relaxing the Primer3 parameter ‘product size’. Specifically, we relax the allowable distance from the target site for one of the primers while keeping the other primer anchored for TAS-opt to a maximum amplicon size of 500 bp. Following Primer3 nomenclature, we call these alternate primer pairs Relaxed-right and Relaxed-left. Arbitrarily, CREPE first attempts to design Relaxed-right primer pairs for the target sites which failed to create TAS-opt pairs; subsequently, it attempts to design and evaluate Relaxed-left primer pairs for any remaining target sites (Figure 1C).

### 3.2. Performance and Storage Needs

CREPE is designed to optimize computational resources while minimizing the overall storage needed for the intermediary and final output files. Anecdotally, across different HPC environments and local installations, CREPE consistently designed and analyzed primer pairs for up to 1000 primer pairs in less than 15 min. To formally test the performance of CREPE, we created an input target list from 1000 randomly chosen ClinVar variants (see Section 2) [31]. We then subdivided this list to generate input lists containing 10, 100, 500, and 1000 target sites and tested their performance on a local computing device. We also assessed the performance of the three components of the CREPE pipeline independently. Primer3 consistently took the least amount of computational time, whereas ISPCR generally took the most. The E-Script generally landed between those two (Figure 2A). We also tested 5000 variants, which is an unlikely scenario for most practical applications. At this scale, the E-Script increased the run time more than expected, suggesting computational bottlenecks that would need to be addressed if more than 1000 variants were routinely required; however, this non-linear increase was mainly dependent on the inclusion of target sites with an outsized number of off-targets. The larger number of tested variants increased the risk of their inclusion. This prevented more general conclusions about processing times for numbers above 1000.

The storage needed for output files from CREPE is minimal compared to the setup requirements, which include a genome reference file, and it increases with the number of input target sites as expected (Figure 2B). Similarly to the processing time, the number of off-targets is variable across target sites and can cause variability.

### 3.3. In Silico Performance of CREPE on 1000 ClinVar Variants

To evaluate CREPE’s ability to perform primer design in potentially disease-relevant genomic regions, we assessed the created TAS primers for all 1000 randomly selected ClinVar database variants [31]. Employing the iterative process described above, CREPE successfully designed primer pairs for 988 of the 1000 target sites (Figure 3A). Of these 988 primer pairs, 860 were deemed acceptable by the E-Script (87.04%; no off-targets or low-quality off-targets) and 128 were rejected (12.96%; high-quality off-targets). Of the 860 acceptable primer pairs, 660 were TAS-opt primer pairs and 200 (23.3%) were Relaxed-right or Relaxed-left primer pairs. This highlights the relative improvement in yield achieved by including the two additional design strategies. The 860 primer pairs could also be split into two groups, namely primer pairs with zero predicted off-targets and primer pairs with at least one predicted low-quality off-target (Figure 3B). None of the primer types were enriched for primer pairs with off-targets, suggesting that there is no compromise in quality employing the iterative approach. Of note, across the 128 rejected primer pairs, 33 (25.8%) were Relaxed-right or Relaxed-left, also suggesting that high-quality off-targets are not relevantly enriched when a TAS-opt primer cannot be designed.

To account for possible bias in our initial selection of 1000 ClinVar variants, we repeated this analysis on 100 additional sets of 1000 input variants randomly sampled from the ClinVar database. Those results were comparable to the initial test set, supporting that the findings in the test set were not due to any sampling bias (Appendix A and Appendix A). Overall, these analyses reveal that CREPE can successfully design and annotate primer pairs and that our TAS-specific iterative approach increases the yield of primers, as approximately one-fifth of the successfully designed primers required the use of relaxed conditions.

### 3.4. Experimental Evaluation of CREPE Performance

In order to extend our validation of CREPE’s primer design performance, we executed three independent experimental TAS runs as follows: (1) 95 randomly selected targets that had no off-targets (‘No-Off’); (2) 96 randomly selected targets that had either no off-targets or those of low quality (91 and 5, respectively; ‘Low-Off’); and (3) 21 randomly selected targets that had off-targets of high quality (‘High-Off’). The exact breakdown of primer types is shown in Table 1.

The TAS experiment was performed with the selected primers on control DNA that was not expected to harbor the selected ClinVar variants. Each dataset was submitted for sequencing as a separate pooled library on different sequencer runs and analyzed separately (see Section 2). We first assessed the background coverage across the genome for regions with at least one read for each experiment and compared it to the distribution of coverage for the predicted amplicons (Figure 4). Considering all targets that had no coverage or coverage indistinguishable from the background across the three experiments, 175/186 (94.1%) targets with no off-targets (Z-Off), 5/5 with low-quality off-targets (LQ-Off), and 19/21 (90.5%) with high-quality off-targets (HQ-Off) were successfully amplified with TAS.

As the amplicons with predicted high-quality off-targets result in the submission of additional irrelevant amplicons, one would predict a relatively reduced on-target coverage for these primers. Indeed, we observed a reduced normalized coverage for these primers compared to the Z-Off primer pairs, although this result was not significant (*p* = 0.0564, Mann–Whitney U between Z-Off and HQ-Off, Figure 5A; Appendix A). To further evaluate the impact of off-targets on sequencing performance, we specifically measured the coverage for the predicted off-targets in the Low-Off and High-Off datasets. Only 1/8 (12.50%) of the low-quality off-targets (LQ-Off) had coverage, with a relatively modest read depth of 519× (Figure 5B). There were 8783 predicted off-targets in the High-Off dataset, with 8407 belonging to a single primer pair (clinvar2_19_19934554_ideal; Appendix A). Due to overlaps of these off-targets, this resulted in 400 off-target regions with an average length of 1016 bp (see Section 2). Across these regions, 108 showed coverage above the background; these off-targets were contributed by 18 of the 21 amplicons with a non-normalized median coverage of 1590× (Appendix A; Appendix A). These results confirmed the predictions by CREPE for primer pairs with low- and high-quality off-targets.

In summary, using our in silico analysis and TAS validation results as an approximation, we estimate that CREPE can create a primer pair with no off-targets or low-quality off-targets that enrich above the background coverage for approximately 81.05% ± 6.6% of the input target sites (Figure 6). This demonstrates that CREPE represents a novel tool that allows for the efficient creation and evaluation of primers for target regions at scale. While the core functionalities leverage existing and trusted tools, we add additional functionalities to improve the user experience and interpretation of the obtained primers [6,7,12]. While aspects of our workflow are automated and provide interpretation, it is easily possible for users to access the results that underlie the provided annotations. As applications differ from each other, off-targets might be an issue at any appreciable level and sequence identity or be negligible if the amplicon sequence itself differs. To further support such decision-making, we also provide information on the content of amplicons and their similarity. For instance, a targeted region might have off-targets that are likely to amplify but are dissimilar in sequence to the on-target region (i.e., low-quality off-targets); this is acceptable for some next-generation sequencing applications but unacceptable for bulk analyses like those provided through Sanger sequencing. Similarly, some predicted off-targets may be the result of primers binding near the target site and creating amplicons overlapping the target amplicon. While those off-targets similarly might not interfere with TAS validation, they would again be incompatible with Sanger sequencing. Finally, our current implementation is optimized for genomic PCR amplifications and does not account for gene or exon boundaries. While this could be addressed partially by customized references and positional information, it currently limits usage for non-genomic PCR applications.

Currently, CREPE requires basic knowledge of the command line language and the ability to run programs within this environment. While we have streamlined the process as much as possible, this might still represent a hurdle for many users. Thus, future goals for CREPE include the development of a GUI that allows for parallel primer design for users with limited experience in text-based user interfaces.

## Figures and Tables

**Figure 1 genes-16-01062-f001:**
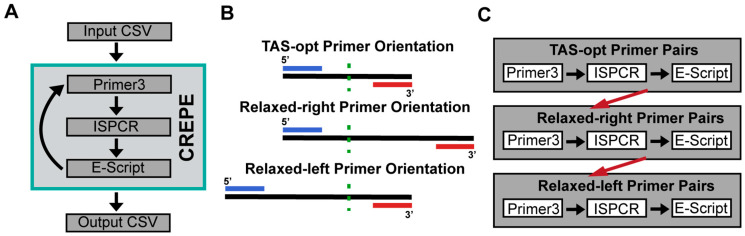
General overview of the CREPE pipeline. (**A**) The CSV-formatted input table containing the target sites is processed for input into Primer3; ISPCR finds off-targets and verifies on-target matches for any designed primer pairs; a custom Python-based evaluation script (E-script) analyzes off-targets, filters out low-quality off-targets, and merges results back to the original input CSV to generate the output CSV. (**B**) Schematics of the primer pair orientations that CREPE allows for TAS experiments. The green dashed line represents the target site (variant of interest), the blue line the forward primer, and the red line the reverse primer. For TAS-opt, the forward and reverse primers are an equal distance from the target site within 60–100 bp. For Relaxed-right, the considered genomic region is extended for the reverse primer to find a viable binding site outside of the area assessed by short-read sequencing. Similarly, for Relaxed-left, the forward primer can be located farther from the target site. The maximum amplicon size for the alternate primer pairs is 500 bp. 5′ and 3′ ends are marked, left and right, respectively, according to Primer3 conventions. (**C**) CREPE designs the TAS-opt primers first. It then attempts to create a Relaxed-right primer pair for any target site that fails to create a TAS-opt primer pair. Lastly, CREPE will attempt to create a Relaxed-left primer pair for any remaining target sites. This iterative process reduces computation time and storage. After completing this iterative process, the final output table is exported in CSV format (see Appendix A).

**Figure 2 genes-16-01062-f002:**
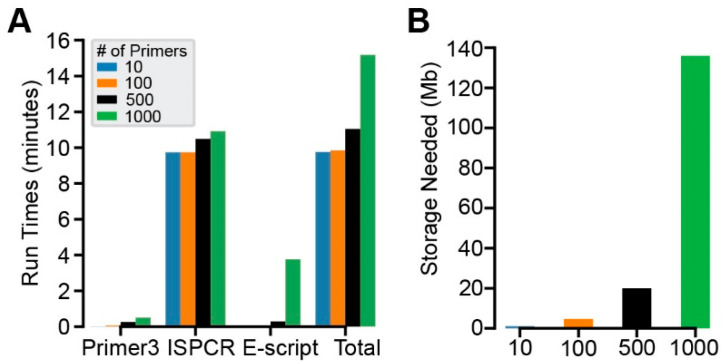
Performance of CREPE. (**A**,**B**). Computation time (**A**) and storage requirements (**B**) for 10, 100, 500, and 1000 target sites. The computation times shown here were obtained from a CREPE instance installed on a local device (see Section 2). Runtimes (minutes): Primer3: 0.007, 0.061, 0.261, and 0.510. ISPCR: 9.733, 9.739, 10.487, and 10.916. Python (minutes): 0.013, 0.038, 0.289, and 3.751. Total: 9.75, 9.83, 11.04, and 15.18. Required storage for output files from CREPE (Mb): 1.0, 4.6, 20.0, and 136.0, respectively, for the different numbers of target sites.

**Figure 3 genes-16-01062-f003:**
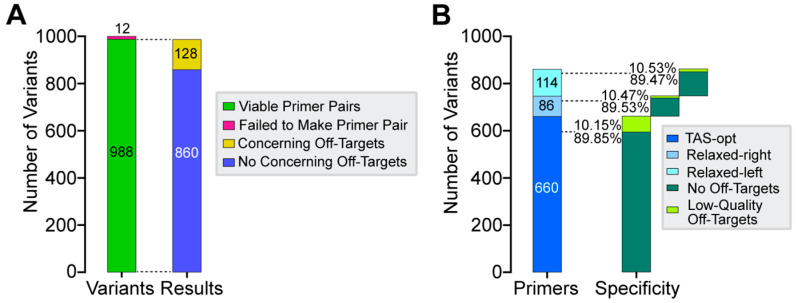
Implementation of CREPE for 1000 ClinVar variants. (**A**) In total, 988 (98.8%) viable primer pairs were created successfully, with 860/988 (87.04%) having no predicted high-quality off-targets (acceptable) and 128/988 (12.96%) having at least one predicted high-quality off-target (rejected). (**B**) Detailed assessment of designed primer type across the 860 primer pairs with no concerning (i.e., high-quality) off-targets. A total of 660 (76.74%) were TAS-opt primer pairs and 200 (23.26%) were alternate primer pairs; 86 of these were Relaxed-right and 114 Relaxed-left. Each primer orientation had a mix of primer pairs with no predicted off-targets or at least one predicted low-quality off-target: 89.85% and 10.15%, respectively, for TAS-opt; 89.53% and 10.47%, respectively, for Relaxed-right; and 89.47% and 10.53%, respectively, for Relaxed-left.

**Figure 4 genes-16-01062-f004:**
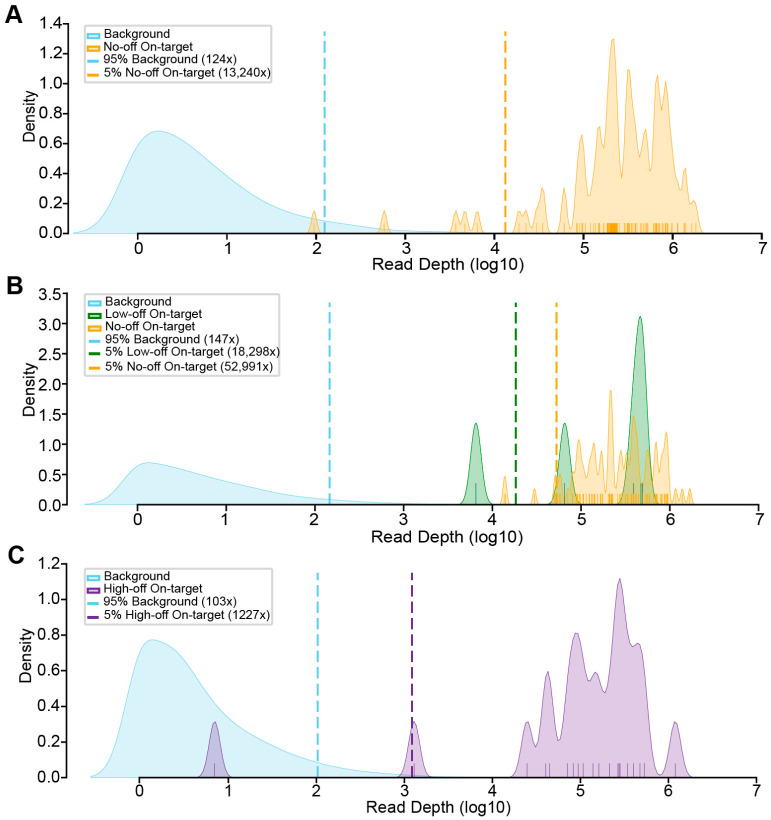
Comparison of background enrichment and on-target coverage. (**A**) KDE plots of the background and on-target coverage distributions for the No-Off dataset. The 95th percentile of the read depth distribution for the background data was 124×. Three targets had no coverage (not shown in plot), and one of the tested primer pairs with detected reads failed to enrich above the background with a read depth of 95×. (**B**) KDE plots of the background and on-target coverage distributions for the Low-Off dataset. The 95th percentile of the read depth distribution for the background data was 147×. Seven targets (all without concerning off-targets) had no coverage (not shown in plot), and all tested primer pairs with detected reads enriched above the background. (**C**) KDE plots of the background and on-target coverage distributions for the High-Off dataset. The 95th percentile of the read depth distribution for the background data was 103×. One target had no coverage (not shown in plot), and one of the tested primer pairs with detected reads failed to enrich above the background with a read depth of 7×. For all panels: 95% Background: 95th percentile of the background read depth distribution for each dataset; 5% On-target: 5th percentile of the on-target read depth distribution for each dataset.

**Figure 5 genes-16-01062-f005:**
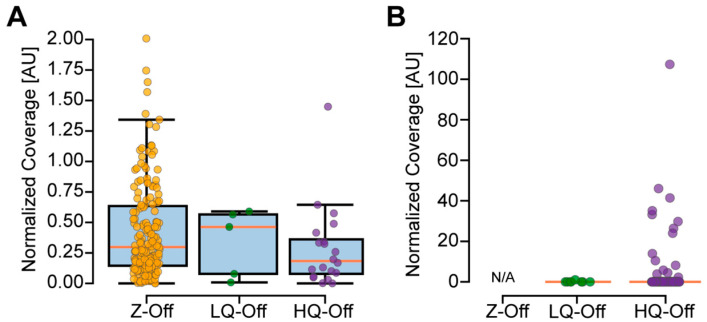
Normalized coverage across primer pairs and their predicted off-targets. (**A**) A total of 176 primer pairs without off-targets (Z-Off) had on-target coverage (non-normalized median = 281,708×), five with low-quality off-targets (LQ-Off) had on-target coverage (non-normalized median = 387,355×), and 20 with high-quality off-targets (HQ-Off) had on-target coverage (non-normalized median = 150,844×). Kruskal–Wallis group test and post hoc Mann–Whitney U test did not find a significant difference despite visible trend in data. (**B**) By definition, there were no predicted off-targets for the Z-Off amplicons. A total of 1/8 (12.50%) of the predicted low-quality off-targets in the Low-Off dataset had coverage (non-normalized coverage of 519×); 226/400 (56.50%) predicted high-quality off-target regions in the High-Off dataset had coverage (non-normalized median = 71×), 108 of which were above background (non-normalized median = 1590×).

**Figure 6 genes-16-01062-f006:**
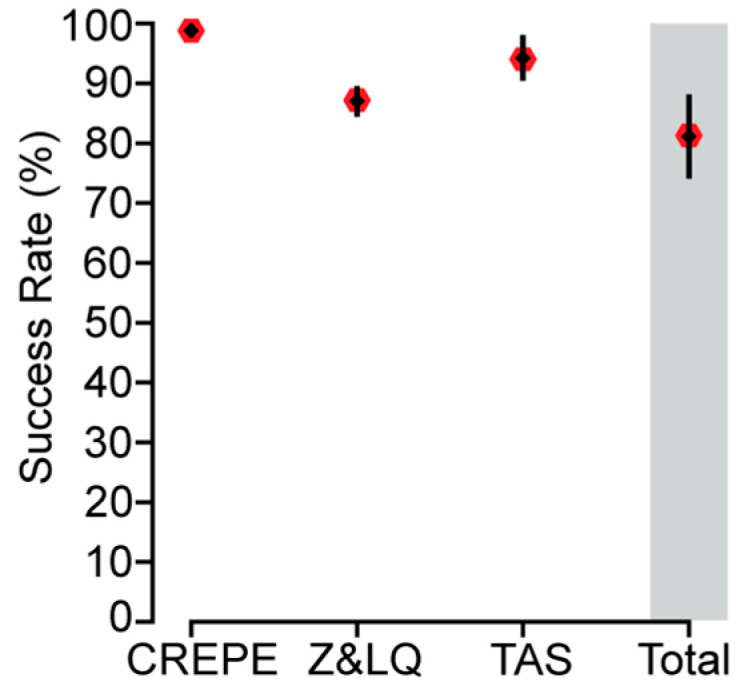
Summary of CREPE performance for primer pairs with no predicted off-targets (Z-Off) or low quality off-targets (LQ-Off). CREPE successfully created 988/1000 (98.8% ± 0.7%) primer pairs. Of those, 860/988 (87.05% ± 2.4%) were predicted to have either no off-targets (Z-Off) or low-quality off-targets (LQ-Off). When tested with TAS validation, 180/191 (94.24% ± 3.5%) enriched the target site above the background. The total is the combination of these three analyses (81.05% ± 6.6%). The gray bar highlights the total.

**Table 1 genes-16-01062-t001:** Description of primer datasets used for TAS validation.

Primer Dataset	TAS-opt	Relaxed-Right	Relaxed-Left	Total
No-Off	76	6	13	95
Low-Off	71	8	17	96
High-Off	16	2	3	21

## Data Availability

https://github.com/martinbreuss/BreussLabPublic/tree/main/CREPE (accessed on 9 September 2025) (DOI:10.5281/zenodo.15230814). Sequencing data is available on the SRA under accession: PRJNA1250513.

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
