# Peer review of "CREPE (CREate Primers and Evaluate): A Computational Tool for Large-Scale Primer Design and Specificity Analysis"

_genes, 2025, doi:10.3390/genes16091062_

Round 1

Reviewer 1 Report

Comments and Suggestions for Authors

The authors present a computational pipeline, CREPE, for high-throughput human targeted-amplicon primer design with user friendly reports. By coupling Primer3 for candidate generation with specificity and off-target checks, CREPE streamlines the process to provide primer design and its evaluation for targeted sites at scale. The tool produces clear per-target reports that reduce manual curation, demonstrate its result in silico, and appears to achieve strong on-target rates in their wet-lab validation. Overall, it fills a useful gap between point-and-click utilities and custom lab scripts for large-scale human TAS primer design, making it reproducible and accessible.

I have a few minor comments regarding the work.
1. The authors mentioned separate PCRs for initial amplification, hence the workflow does not support multiplex reactions. Adding an optional pooling/compatibility optimization could strengthen CREPE and broaden its applicability.

2. For figure 2, the authors reported that the E-script increased the run time more than expected for 5k variants, and that this was dependent on off-target counts. Can the authors comment on the off-target counts for the variants (10, 100, 500, 1000) plotted? Did the author mean scaling up to more than 1k variants resulted in higher chances of variants getting more off-target counts, hence the increase in time? This was unclear to me.

3. Lastly, this may be out of scope, there are other exon or gene specific primer design tools, while the CREPE is not gene-specific. It may be helpful to comment on that in background.

Author Response

The authors present a computational pipeline, CREPE, for high-throughput human targeted-amplicon primer design with user friendly reports. By coupling Primer3 for candidate generation with specificity and off-target checks, CREPE streamlines the process to provide primer design and its evaluation for targeted sites at scale. The tool produces clear per-target reports that reduce manual curation, demonstrate its result in silico, and appears to achieve strong on-target rates in their wet-lab validation. Overall, it fills a useful gap between point-and-click utilities and custom lab scripts for large-scale human TAS primer design, making it reproducible and accessible.

Response: We thank the reviewer for the overall positive assessment of our proposed bioinformatic tool and its intended use-cases. We provide answers to the three raised points below.

I have a few minor comments regarding the work.

Comment 1. The authors mentioned separate PCRs for initial amplification, hence the workflow does not support multiplex reactions. Adding an optional pooling/compatibility optimization could strengthen CREPE and broaden its applicability.

Response 1. We agree that the ability to multiplex PCRs is highly valuable across a range of applications. Several of the more complex tools cited in our introduction are able to provide this functionality and require more complex analyses of the possible interactions of primers with each other and the potential off-targets that result from mixed primer pairs. As a consequence, we believe that this application is out of scope for our proposed tool, and we decided to focus on our current application that is of critical importance for many other applications but might be overcomplicated by the use of pooling-optimized tools.

Comment 2. For figure 2, the authors reported that the E-script increased the run time more than expected for 5k variants, and that this was dependent on off-target counts. Can the authors comment on the off-target counts for the variants (10, 100, 500, 1000) plotted? Did the author mean scaling up to more than 1k variants resulted in higher chances of variants getting more off-target counts, hence the increase in time? This was unclear to me.

Response 2. We apologize that we did not convey this point more clearly, but the reviewer is correct: when scaling the primer design to larger numbers, the risk of including a site with a significantly higher number of off-targets increases. These high off-target sites are mainly responsible for non-linear increases in processing time. We have adjusted our language to specifically mention this potential limitation.

“At this scale, the E-Script increased the run time more than expected, suggesting computational bottlenecks that would need to be addressed if more than 1,000 variants were routinely required; however, this non-linear increase was mainly dependent on the inclusion of target sites with an outsized number of off-targets. The larger number of tested variants increased the risk of their inclusion. This prevented more general conclusions about processing times for numbers above 1,000.”

Comment 3. Lastly, this may be out of scope, there are other exon or gene specific primer design tools, while the CREPE is not gene-specific. It may be helpful to comment on that in background.

Response 3. We agree that this is a current limitation of CREPE that restricts its usage outside of genomic PCR applications. We have added this limitation to our discussion.

“Finally, our current implementation is optimized for genomic PCR amplifications and does not account for gene or exon boundaries. While this could be addressed partially by customized references and positional information, it currently limits usage for non-genomic PCR applications.”

Reviewer 2 Report

Comments and Suggestions for Authors

The manuscript by Pitsch and colleagues presents a bioinformatics tool called "CREPE" (CREate Primers and Evaluate) for large-scale PCR primer design. Specifically, this tool integrated two software packages (Primer3 and ISPCR) into a pipeline to steamline the design and specificity evaluation of primers with arbitrary number of target sites. For each target site, the software provides likelihood-based score to measure the probability of off-target amplification. To validate the performance, the authors tested CREPE on a paired-end Illumina sequencing platform and reported good yield of the primers with high specificity.

Comments and Suggestions:

  1. The authors should be commended for making their code open-source and available at GitHub. I checked out the code repository and I was able to run the example task. I find that the code organization and the documentation can be better polished. For example, the file crepe_download.tar.gz contains contents (CREPE_v1.02.py, clinvar10_demo.csv, env.yml) to the root directory (BreussLabPublic/CREPE/). If the authors hope to separate the package under development and the package for download, it is usually recommended to create a release. In addition, "3. If you have an issue with the tar file, Git clone this directory" (BreussLabPublic/CREPE/README.md, Line 42) sounds confusing as the goal to do so is unclear. Finally, for the output table, the authors are recommended to document the meaning of the key columns (e.g. TAS-opt, primer_count, etc.) so that end-users can grasp this tool quickly without referring to the article.

  2. Figure 2A, a bar plot should be more appropriate here as the x-axis groups are parts, not trends, and there is no point connecting them into lines.

  3. Line 324-426, "…our TAS-specific iterative approach increases the yield of primers". How does it compare to the state-of-the-art performance? Is there any way to show it?

Author Response

The manuscript by Pitsch and colleagues presents a bioinformatics tool called "CREPE" (CREate Primers and Evaluate) for large-scale PCR primer design. Specifically, this tool integrated two software packages (Primer3 and ISPCR) into a pipeline to steamline the design and specificity evaluation of primers with arbitrary number of target sites. For each target site, the software provides likelihood-based score to measure the probability of off-target amplification. To validate the performance, the authors tested CREPE on a paired-end Illumina sequencing platform and reported good yield of the primers with high specificity.

Response. We thank the reviewer for their comments and suggestions to improve the tool distribution, visualization of data, and clarification. Please see below for specific comments.

Comments and Suggestions:

Comment 1. The authors should be commended for making their code open-source and available at GitHub. I checked out the code repository and I was able to run the example task. I find that the code organization and the documentation can be better polished. For example, the file crepe_download.tar.gz contains contents (CREPE_v1.02.py, clinvar10_demo.csv, env.yml) to the root directory (BreussLabPublic/CREPE/). If the authors hope to separate the package under development and the package for download, it is usually recommended to create a release. In addition, "3. If you have an issue with the tar file, Git clone this directory" (BreussLabPublic/CREPE/README.md, Line 42) sounds confusing as the goal to do so is unclear. Finally, for the output table, the authors are recommended to document the meaning of the key columns (e.g. TAS-opt, primer_count, etc.) so that end-users can grasp this tool quickly without referring to the article.

Response 1. We agree with the suggested re-organization of the GitHub presentation to improve usability. To this end we have done the following: we created a versioned subdirectory that contains all relevant files and a ‘current_release’ directory to prepare for future releases. We further removed the quoted step that made the instructions more confusing and focused on providing only the instructions related to the tar file installation. Finally, we have also added the requested key columns to the README.md.

Comment 2. Figure 2A, a bar plot should be more appropriate here as the x-axis groups are parts, not trends, and there is no point connecting them into lines.

Response 2. We agree with the reviewers suggestion and we have exchanged the panel with bar plots.

Comment 3. Line 324-426, "…our TAS-specific iterative approach increases the yield of primers". How does it compare to the state-of-the-art performance? Is there any way to show it?

Response 3. We apologize for the lack of clarity. This sentence was intended to indicate that the use of relaxed conditions was required for 23.3% of the successful primers within in our pipeline, compared to only considering the TAS-optimized primer pairs. We have amended this sentence to include such a statement:

“Overall, these analyses reveal that CREPE can successfully design and annotate primer pairs, and that our TAS-specific iterative approach increases the yield of primers, as approximately one-fifth of the successfully designed primers required the use of relaxed conditions.”